# Electrospun Medicated Nanofibers for Wound Healing: Review

**DOI:** 10.3390/membranes11100770

**Published:** 2021-10-09

**Authors:** Xinkuan Liu, Haixia Xu, Mingxin Zhang, Deng-Guang Yu

**Affiliations:** 1School of Materials Science and Engineering, University of Shanghai for Science and Technology, Shanghai 200093, China; 193742716@st.usst.edu.cn (H.X.); 203613006@st.usst.edu.cn (M.Z.); ydg017@usst.edu.cn (D.-G.Y.); 2Shanghai Engineering Technology Research Center for High-Performance Medical Device Materials, Shanghai 200093, China

**Keywords:** wound dressing, electrospinning, nanostructure, nanocomposite

## Abstract

With the increasing demand for wound care and treatment worldwide, traditional dressings have been unable to meet the needs of the existing market due to their limited antibacterial properties and other defects. Electrospinning technology has attracted more and more researchers’ attention as a simple and versatile manufacturing method. The electrospun nanofiber membrane has a unique structure and biological function similar to the extracellular matrix (ECM), and is considered an advanced wound dressing. They have significant potential in encapsulating and delivering active substances that promote wound healing. This article first discusses the common types of wound dressing, and then summarizes the development of electrospun fiber preparation technology. Finally, the polymers and common biologically active substances used in electrospinning wound dressings are summarized, and portable electrospinning equipment is also discussed. Additionally, future research needs are put forward.

## 1. Introduction

Skin is the largest important organ of the human body and the first barrier against external pathogens [1]. However, external mechanical forces, surgical operations, burns, chemical injuries, and ulcers from certain chronic diseases can cause varying degrees of damage to the skin [2]. Wound healing is a complicated and dynamic process of tissue regeneration, mainly composed of four stages: hemostasis, inflammation, proliferation, and remodeling [3]. Although the skin can undergo a certain degree of spontaneous repair, bacterial infection has always been the main reason hindering wound healing. For an infected wound, it will not only disrupt the normal healing process, but also cause the wound tissue to be deformed, causing great pain to the patient [4].

Wound dressings play an essential role in wound healing management. They protect the wound from external risk factors, and speed up the healing process [5]. On the basis of the mechanism of wound healing, an ideal wound dressing ought to have the accompanying attributes: (1) absorb excess exudate; (2) protect the wound from microbial infection; (3) maintain a moist healing environment at the wound site; (4) facilitate gas exchange; (5) non-toxic, biocompatible, and degradable; (6) does not adhere to the wound, easy to replace and remove; (7) promote angiogenesis and tissue regeneration [6,7,8]. Different wound needs should be integrated when choosing wound dressings. So far, the common dressings on the market mainly include film [9], foam [10], sponge [11], hydrogel [12,13], and nanofiber membrane [14,15]. Among these materials, the unique structure of the small pore size and high porosity of the nanofiber membrane can protect the wound from pathogen infection and ensure the free transportation of gas and liquid molecules. At the same time, a large amount of research has been carried out, combining the adjustable characteristics of physical and mechanical properties to make it stand out among biomaterials [16,17].

So far, methods such as drawing [18], self-assembly [19], phase separation [20] and template synthesis [21] have been used to prepare nanofibers. However, they have disadvantages such as high cost, time-consuming and low efficiency. Therefore, simple and practical electrospinning technology is widely used to manufacture fibers with diameters in the nanometer or micrometer range [22]. Electrospun nanofiber membranes represent a new class of materials. Because of their high surface-to-volume ratio, high microporosity and versatility, they can be used in various biomedical applications [23], such as tissue engineering scaffolds [24,25], drug delivery [26,27,28] and wound dressings [29,30]. Nanofiber wound dressings prepared by electrospinning technology have many advantages. First, the structure and biological function are similar to the natural extracellular matrix (ECM), which provides an ideal microenvironment for cell adhesion, proliferation, migration and differentiation [31,32]. Secondly, the polymer matrix used for electrospinning can simultaneously combine the biocompatibility of natural polymers and the reliable mechanical strength of synthetic polymers [33]. Furthermore, the nanofiber membrane’s wide surface area and porous structure can be effectively loaded with various biologically active ingredients, including antibacterial drugs, inorganic nanoparticles, vitamins, growth factors and Chinese herbal extracts. The rate and time of drug release are controlled by adjusting the fiber structure and morphological size, thereby promoting effective healing of the wound site [34]. Therefore, electrospun nanofibers show great potential in the preparation of advanced bioactive wound dressings.

In recent years, on the “Web of Science” platform, the subject of “Wound dressing” and “Electrospun wound dressing” has been searched among the literature. The statistical results are shown in Figure 1. The number of relevant documents corresponding to the two topics has shown a substantial increase year by year. Among them, the literature with the theme of “Wound dressing” has maintained thousands of articles every year from 2010 to the end of 2020, indicating that wound dressing has become a hot research topic in recent years. At the same time, the number of documents retrieved with the theme of “Electrospun wound dressing” has maintained more than a hundred articles in the past five years, indicating that electrospun nanofibers are developing into a new type of wound dressing with broad application prospects. This article reviews the research progress and application prospects of electrospun medical nanofibers used in wound dressings. The new strategy of electrospinning technology for preparing nanofiber wound dressings is described in detail.

## 2. Wound and Wound Dressing

### 2.1. Wounds Classification

Wounds are defined as skin deformities or tissue discontinuities brought about by physical or thermal injury, or underlying ailments [35]. Given the nature and duration of the healing process, wounds are usually divided into acute and chronic types [36]. Acute wounds mainly include mechanical injuries, chemical injuries, surface burns and surgical wounds, etc. The healing process follows the normal wound healing cycle [37,38,39]. However, chronic wounds refer to those cannot go through an orderly healing process and have been open for more than one month. The causes of chronic wounds vary, and are mainly related to certain specific diseases (such as diabetes). They are notorious for the terrible incidence of ulcers, and they are susceptible to infection by inflammatory bacteria that affect wound repair [40,41]. Globally, chronic wounds impose a heavy burden on patients and healthcare systems [42].

### 2.2. Types of Wound Dressing

In 1962, Dr. Jorge Winter of the University of London put forward the “moist healing environment theory” first, and related studies confirmed that a moist environment will speed up the wound healing process [43]. In recent years, the theory of moist healing has received extensive consideration. The U.S. Food and Drug Administration (FDA) pointed out in an industry guide issued in August 2000 that one of the standard methods of wound treatment is to maintain a moist environment on the wound surface [44]. With the in-depth study of wound healing, the types of wound treatment and dressings are constantly improving and developing [45]. Wound dressings are classified into traditional wound dressing, modern wound dressing and bioactive wound dressing according to their functional properties and wound origin. Table 1 classifies and summarizes wound dressings based on their functions.

#### 2.2.1. Traditional Wound Dressing

Traditional wound dressings mainly use gauze, lint and bandages as dressings, which are the most widely used wound dressings. However, it does not absorb and drain the exudate from the wound site smoothly during use, and cannot maintain a moist healing environment. The porous structure on the surface cannot prevent the invasion of external bacteria and will promote the migration of bacteria to the wound site in a humid environment. It is easy to adhere to the wound exudate during use, so it will cause secondary damage to the skin when replaced [53].

#### 2.2.2. Modern Wound Dressing

Compared with traditional dressings, modern wound dressings have better biocompatibility, moisture retention and degradability [54]. Standard high-end modern dressings include film, foam, hydrocolloid, hydrogel and alginate dressings, etc., which provide a more effective healing effect than traditional dressings [47]. The film is a light, thin and elastic transparent polyurethane or synthetic polymer adhesive. The transparent surface can permeate not only water vapor and oxygen, but is also convenient to observe the healing of the wound [55]. Foam dressings are usually composed of a hydrophilic polymer layer and a hydrophobic polyurethane absorbent layer. The polymer layer is permeable to gas and semi-permeable to fluids. The inner layer structure is used to absorb exudate, and the outer layer is used to prevent the wound from drying out and prevent it from being invaded by bacteria [56]. Hydrocolloid dressing is a new type of wound dressing widely used in clinical practice, which is made of water or glycerin-based polymer materials. After absorbing the liquid, it will swell to form a gel, and the dressing will become soft [57]. Hydrogel dressings are cross-linked polymer networks made of carbohydrate-based materials, which can be made into various thicknesses and are widely used in drug delivery and tissue repair [58]. Alginate dressings are a dressing prepared with different types of seaweed or algae polysaccharides. It has a high fluid absorption capacity and promotes autolytic debridement to soften and eliminate necrotic tissue. It is often used for chronic wounds with high exudate fluid, deep burns and surgical wounds that require hemostasis [59].

#### 2.2.3. Bioactive Wound Dressing

As a new type of wound dressing, bioactive wound dressings are used to expedite the healing of various types of wounds [60]. This dressing is made of various polymers, such as gelatin, silk fibroin, chitosan, alginate, etc., and is applied in the form of foam, sponge, film, hydrogel and nanofiber membrane (Figure 2) [6,61]. Biologically active substances such as antibacterial drugs, growth factors, nanoparticles and natural products are added to obtain antibacterial activity, promote fibroblast activity and endothelial cell migration [62]. In short, the development of functionalized bioactive dressings is a crucial step in developing wound dressings.

## 3. Electrospinning Technology

### 3.1. Introduction to Electrospinning Technology

Electrospinning technology, as a superfine fiber preparation technology, has experienced hundreds of years of development [63]. The electrostatic spinning device is mainly composed of four parts: a high-voltage generator, a fluid driver, a spinneret and a collection device [64]. In the electrospinning process, the initial electrospinning fluid gradually changes its morphology after the voltage is applied, until it reaches the critical voltage shape into a Taylor cone. When the liquid jet stretches over a certain distance, it enters the bending and whiplash stage. With the solvent volatilization, the jet is stretched to micrometers or even tens of nanometers, finally solidified and deposited on the collector to form nanofiber [65,66]. On the basis of this principle, the electrospinning process can be adjusted by system parameters (polymer type, molecular weight, viscosity, conductivity of the solution, surface tension), process parameters (voltage, flow rate, receiving distance) and environmental parameters (humidity, temperature) to change the morphology and size of nanofibers [67,68]. As a simple, top-down one-step preparation method, electrospinning technology produces nanofibers with small pore size, high porosity and a structure similar to ECM. Therefore, it has received extensive attention from researchers and used to prepare functionalized nanofibers for applications in biomedicine and other fields [69,70,71]. At the same time, the electrospinning technology is continuously upgraded and optimized. As shown in Figure 3, it has gradually developed into single fluid electrospinning (blend electrospinning and emulsion electrospinning), double-fluid electrospinning (coaxial electrospinning and side-by-side electrospinning) and multifluid electrospinning (triaxial electrospinning and other multifluid electrospinning).

### 3.2. Single Fluid Electrospinning

#### 3.2.1. Blend Electrospinning

Blend electrospinning is the most common method for preparing blended nanofibers. A suitable solvent is usually selected in the preparation process, which can dissolve the polymer used and the drug added. If the polymer-dissolving solvent cannot dissolve the drug, it is possible to add a small amount of the drug-soluble solvent to dissolve them together [73]. If the dissolvability of the drug in the polymer solution is low, a large amount of the drug will be present on the surface of the prepared fiber. As a result, an explosive release of drugs occurs, which is also the main weakness of hybrid electrospinning [74].

#### 3.2.2. Emulsion Electrospinning

Emulsion electrospinning is a simple and standard method for preparing nanofibers with a core–shell structure, which could effectively load drugs [75]. Different from the blend electrospinning mentioned above, the hydrophilic drugs in emulsion electrospinning are usually dissolved in the water phase and then diffused into the oil phase containing surfactants/emulsifiers. The water/oil emulsion obtained after electrospinning forms nanofibers with a core–shell structure in which the drug is included in the core [76]. Therefore, it shows superior performance in the properties of encapsulating biologically active substances, e.g., proteins and drugs [77]. Zhan et al. [78] reported that using polyvinyl alcohol (PVA) and polyacrylic acid (PAA) as polymers, electrospun nanofibers loaded with tangeretin (Tan) were prepared by the emulsion electrospinning method. In vitro drug release results show that PVA/PAA/Tan nanofibers have longer-lasting release characteristics and a lower initial burst release rate than pure Tan emulsion. This work provides a promising technology for the preparation of insoluble drug release and drug delivery systems.

### 3.3. Double-Fluid Electrospinning

#### 3.3.1. Coaxial Electrospinning

Among the various multifluid electrospinning techniques, coaxial electrospinning is the most basic one, which is widely used to manufacture core–sheath nanofibers [79]. In the core–sheath nanofibers, if a drug-loaded hydrophilic polymer is selected as the core solution, voids will be formed in the sheath fiber after it is dissolved. At this time, the hydrophobic shell serves as an outer protective layer to prevent initial burst release. By loading different forms of sheath polymer, two different release forms can be achieved to control the release of drugs [80]. Therefore, coaxial electrospinning is considered one of the most significant breakthroughs in the field of drug-sustained release [81,82]. Yan et al. [83] successfully prepared PH-sensitive PVA/polycaprolactone (PCL) core–shell nanofibers using coaxial electrospinning technology, achieving sustained PH-responsive release of the anticancer drug doxorubicin (DOX).

With the in-depth study of electrospinning technology, researchers have discovered that the traditional coaxial electrospinning technology has the problem of poor ability to manage the morphology of the fiber. At the same time, more and more research is devoted to treating unspinnable fluids in multifluid work. If they are used as a sheath liquid and work with a spinnable core fluid, they can be converted into nanofibers [84]. In 2010, Yu et al. [85] reported the use of pure solvents (unspinnable fluid) as the sheath working fluid to conduct electrospinning experiments, which effectively prevented the clogging of the spinneret during the spinning process; high-quality nanofibers with smooth surfaces were prepared. Later, the coaxial/triaxial process using unspinnable fluid as the sheath fluid was called modified coaxial/triaxial electrospinning.

#### 3.3.2. Side by Side Electrospinning

In 2003, Gupta and Wilkes first reported the use of side-by-side electrospinning to prepare Janus nanofibers [86]. The Janus structure is one of the most basic structures. Different from the traditional core–sheath structure, its two chambers are separated from each other and both are in contact with the surrounding environment. In side-by-side electrospinning, nanofibers with different properties can be prepared by designing the structure of the spinneret and adjusting the electrospinning parameters [87,88,89]. Zheng et al. conducted a study on the effect of the bilateral drug loading of electrospun Janus nanofibers on the two-phase controlled release of the anticancer drug tamoxifen (TAM), and described the related mechanisms. It can be found from the drug release curve that different polymer components, different two-compartment structures and shapes have a significant impact on the in vitro release characteristics of TAM. They have also been proven to be important elements in the design of functional nanomaterials [90].

### 3.4. Multifluid Electrospinning

#### 3.4.1. Triaxial Electrospinning

With the rapid development of nanotechnology, researchers are not satisfied with the development and application of secondary structures. They have begun to devote themselves to the research of triaxial and even multiaxial electrospinning to meet the needs of different fields [91]. Naveen et al. [92] used poly (lactic-co-glycolic-acid) (PLGA) as the sheath fluid, gelatin as the intermediate layer, PCL as the core layer, Rhodamine B (RhB) and Bovine Serum Albumin (BSA) was loaded into the outer and intermediate layer solutions, respectively, a triaxial electrospinning was performed. The prepared nanofibers show excellent mechanical properties and dual drug release capabilities.

Later, Ding et al. [93] used PH-sensitive polymer Eudragit S100 (ES100) as a matrix, loaded with a model drug aspirin, prepared core–shell nanofibers using a modified triaxial electrospinning technology. Compared with traditional single fluid electrospun nanofibers, in vitro drug release exhibits a longer-term sustained release of aspirin, avoiding cytotoxicity caused by short-term excessive drug release. Subsequently, Zhao et al. followed the technology of the research group and used the prepared nanofiber membranes loaded with functional particles for the research of removing antibiotics from water bodies [94]. Wang et al. demonstrated the construction of drug libraries into core–shell nanofibers through modified triaxial electrospinning technology. Cellulose acetate (CA) was used as the polymer, the model drug acyclovir (ACY) prolonged the sustained release time during in vitro release study. The core–shell structure and uneven drug distribution make it show an excellent structure-performance relationship [95]. Therefore, core–shell nanofibers produced through coaxial and modified triaxial electrospinning techniques deliver various strategies for constructing new functional nanomaterials.

#### 3.4.2. Other Multifluid Electrospinning

In multifluid electrospinning, in addition to the widely used core–shell structure and Janus structure, other multifluid electrospinning has also been carried out to study nanofibers with more complex structures. Correspondingly, processes such as quad-fluid coaxial electrospinning, trifluid side-by-side electrospinning and coaxial electrospinning with a side-by-side core have emerged [96]. Zhang et al. [97] reported a nanofiber system prepared by the quadriaxial electrospinning method. Gelatin was used to construct the second outermost and innermost layer, and polycaprolactone was used to build the second innermost and outermost layer. The antibacterial drug moxifloxacin (MXF) is loaded in different layers to achieve different controlled releases. This structure shows broad application prospects in tissue engineering.

## 4. Electrospun Nanofibers in Wound Dressing

Nanofibers prepared by electrospinning technology show excellent properties in promoting wound healing. Their microstructure is highly fitted to the human body ECM structure, which is conducive to cell growth, proliferation and adhesion [31,98]. At the same time, the high permeability and absorption rate can absorb the exudate formed on the wound surface and maintain a moist healing environment. In addition, the large surface area benefits loading and transporting bioactive ingredients such as drugs and growth factors [34,99]. Therefore, electrospun nanofiber materials are considered to be the ideal choice for wound dressings.

### 4.1. Polymer in Electrospun Wound Dressing

At present, there are hundreds of polymers that can be successfully used to prepare drug carriers by electrospinning. In related research on electrospinning wound dressings, both natural and synthetic polymers have been widely used. Figure 4 simply classifies and summarizes the common polymers in electrospun wound dressings.

#### 4.1.1. Natural Polymer

Natural polymers have non-toxic, biocompatibility and biodegradability properties, especially their own advantages such as antibacterial properties, making them popular in electrospinning wound dressings [100]. The following are several typical natural polymers that are used more frequently.

Gelatin, a natural protein biopolymer, is obtained by hydrolyzing part of collagen and is generally pale-yellow translucent particles or flakes. The products are abundant and cheap, and soluble in hot water, glycerin and acetic acid. It has good biocompatibility and biodegradability and is usually used in combination with synthetic polymer PCL to obtain better performance [101].

Chitosan (CS) is a derivative of chitin through deacetylation. It is a polysaccharide with a unique structure and superior functions [102]. It is soluble in aqueous solutions of organic acids such as formic acid and acetic acid, making them highly viscous and facilitating electrospinning [103]. Due to the amino activity on the backbone and the ductility of the hydroxyl group, it has potential applications in the fields of pharmacy, food packaging and wound treatment [104]. Figure 5A shows chitosan extracted from natural sources, developed into electrospun fiber membrane and its application in wound healing. Figure 5B shows a schematic diagram of the wound healing process of a drug-loaded chitosan dressing, which is designed as a physical barrier to protect the wound from microbial invasion and support the migration and differentiation of fibroblasts. Xia et al. [105] used a simple one-step electrospinning method to prepare transparent chitosan-coated cellulose membranes (CM-CS), and studied their cytotoxicity and antibacterial activity against *Staphylococcus aureus* (*S. aureus*) and *Escherichia coli* (*E. coli*). The wound healing effect was evaluated by constructing a wound healing model in mice.

Silk fibroin (SF) is a protein with a complex structure and has been used as a textile and sewing material for decades [106]. Many types of it can be obtained from bees, spiders, wasps, lacewings and silkworms (Figure 5C). Among them, SF extracted from silkworm cocoons is used in the field of biomedicine because of its outstanding mechanical properties, biocompatibility, biodegradability, flexibility, water vapor permeability and slight antibacterial properties. In particular, it has become an excellent candidate for wound dressing applications [107]. Hadisi et al. [108] used hyaluronic acid (HA) as the sheath spinning solution and SF/zinc oxide (ZnO) as the core to prepare nanofibers with a core–sheath structure. The antibacterial mechanism of ZnO was explored through in vitro antibacterial tests, and the healing results of burn wounds in rats indicated that the dressing has a certain burn treatment effect (Figure 5D). Other natural polymer materials commonly used for electrospinning wound dressings include alginate [109], collagen and cellulose.

#### 4.1.2. Synthetic Polymer

There are more types of synthetic polymers applied in electrospinning than natural polymers, and related research on wound dressings is also abundant. Such polymers have excellent mechanical properties, thermal stability and spinnability, and are widely used as carrier materials for electrospinning wound dressings. The following summarizes several synthetic polymers that are more common in the research of wound dressings.

Polyvinylpyrrolidone (PVP) is a hydrophilic polymer prepared from the monomer vinylpyrrolidone (NVP) through bulk polymerization and solution polymerization. PVP is soluble in water and most organic solvents, and has low toxicity. Due to its excellent biocompatibility, it has been widely used in biomedicine and has become one of the three new drug dressings advocated internationally [110]. Chinatangkul et al. [111] mixed shellac (SHL) with PVP, added Monolaurin (ML) and prepared nanofibers by blending electrospinning technology. It shows excellent inhibitory activity against *S. aureus* and *Candida albicans*, indicating that the electrospun nanofiber is suitable for wound treatment.

PCL is a polymer obtained by ring-opening polymerization of ε-caprolactone, which is a material approved by the FDA. It is a white particle and exists in a semi-crystalline state with a low melting point. Because of its outstanding spinnability and mechanical strength, is being widely applied in wound dressings [112]. He et al. [113] prepared a series of nanofibers with antibacterial, anti-oxidation, tensile properties and electrical activity by electrospinning PCL and CS-grafted polyaniline (QCSP) polymer solution (Figure 6A). In the mouse full-thickness wound defect model, the wound healing process is significantly accelerated, showing broad application prospects.

PVA is a white and odorless polymer. It is soluble in water above 95 °C. In addition to being an important chemical raw material, PVA has also drawn in much consideration in the field of medicine. At present, it has a broad scope of utilizations in artificial joints and wound dressings [114]. Ali et al. [115] used a PVA solution to prepare nanofiber mats with Nigella sativa extract deposited by electrospinning. Wound healing analysis results show that the developed PVA-Nigella sativa nanofibrous mat has good wound healing properties and a short recovery time.

#### 4.1.3. Combination of Natural and Synthetic Polymers

On the basis of the benefits of easy degradation and excellent biocompatibility of natural polymers, and the controllability and reliable mechanical strength of synthetic polymers, combining the two types of fibers is the choice of many researchers [116]. Ramalingam et al. [117] reported using PCL/gelatin as a polymer carrier to prepare a coaxial electrospun core–sheath nanofiber membrane, containing antibiotic minocycline and herbal extracts as a multifunctional scaffold for the treatment of secondary burns. The core–sheath structure maintains the sustained release of biologically active ingredients. In addition to showing strong antibacterial activity, it also promotes the proliferation and diffusion of skin cells. Improved collagen tissue can be observed through the second-degree burn model of pigskin. Figure 6B shows the excellent properties of the prepared PCL/gelatin core–sheath structure mat. Zou et al. [118] reported the preparation of PVA/CS nanofibers with carboxymethyl CS nanoparticles, which encapsulated the antimicrobial peptide OH-CATH30. Results showed that prepared nanofibers contain antibacterial properties and can promote wound healing. Khan et al. [119] successfully encapsulated oregano oil in Poly (L-lactide-co-caprolactone) (PLCL)/SF nanofiber membranes. Cytotoxicity and antibacterial tests showed high biocompatibility and antibacterial activity, respectively. At the same time, it can also improve the quality of wound healing and could be used as a potential wound dressing.

### 4.2. Bioactive Ingredients in Electrospun Wound Dressing

Another important advantage of electrospinning to prepare nanofiber wound dressings is that it can load a variety of biologically active ingredients to prepare functionalized products. At present, to improve the antibacterial properties of dressings, commonly used active substances include antibiotic drugs (ciprofloxacin (CIP), curcumin, metronidazole, tetracycline, gentamicin and diclofenac), inorganic nanoparticles (nanosilver particles (AgNP), ZnO, titanium dioxide (TiO_2_), cerium oxide (CeO_2_)), natural substances (honey, essential oils, chitosan) and growth factors [120,121,122,123].

Augustine et al. [124] reported the development of a new type of nCeO_2_, which contains electrospun poly (3-hydroxybutyrate-co-3-hydroxy valerate) (PHBV) membrane. In vivo wound healing studies in diabetic rats confirmed that PHBV membranes mixed with 1% nCeO_2_ showed perfect cell compatibility, and could be used as promising biomaterials to treat diabetic wound healing (Figure 7A). Yang et al. [125] used the side-by-side electrospinning process to prepare Janus nanofibers containing CIP and AgNP as the polymer matrix, and studied their effects on wound healing. The antibacterial effect in the process provides a new idea for the preparation of new antibacterial wound dressings. Jafari et al. [126] prepared a bilayer nanofiber scaffold based on PCL and gelatin. The top layer contains amoxicillin, and the bottom layer contains n-ZnO to accelerate wound healing. In vitro release test showed the sustained release of amoxicillin. Analysis of wound healing in rats showed that the scaffold improved the shrinkage rate of the wound, enhanced the deposition of collagen and reduced the formation of scars. All results and findings indicate that prefabricated stents can be a promising alternative method for treating skin injuries. Figure 7B shows the characterization analysis of the prepared bilayer nanofiber scaffold. Table 2 summarizes the common polymers and active substances in electrospinning wound dressings.

### 4.3. In Situ Electrospinning in Wound Dressing

Compared with ordinary electrospinning, in situ electrospinning is more convenient and comfortable to use, and can better cover wounds. Simultaneously, dressings can be customized according to patient needs [147]. Qin et al. [148] used a portable electrospinning device to prepare electrospun Zein/Clove essential oil nanofiber. In vitro experiments have observed good biocompatibility and antibacterial effects. In the mouse wound model, it can be observed that the prepared Zein/CEO nanofiber membrane promotes wound healing. Figure 8A shows the diagram of its preparation. Yue et al. [149] used ethanol-soluble polyurethane (EPU) and Fluorinated polyurethane (FPU) as polymer carriers and used a customized electrospinning device to prepare thymol-loaded nanofiber membranes (Figure 8B(a)). The results indicate that the membrane has good breathable, waterproof performance and excellent antibacterial activity (Figure 8B(b)), providing a promising strategy for developing portable electrospinning devices.

### 4.4. Application of Electrospinning Technology in Other Fields

In recent years, the advantages of electrospinning have attracted more and more attention. With the continuous research of related scholars, the application of electrospinning nanofibers has become more and more extensive. In addition to playing a role in the field of biomedicine (drug delivery [150,151,152], tissue engineering [153] and wound dressings), it also plays a pivotal position in environmental protection (air filtration, water treatment), energy and chemical industries (light-emitting device, solar cell and supercapacitor) and other fields [154,155]. Fiber materials with unique structures and characteristics arranged by electrospinning have been generally utilized in different fields (Figure 9). Combining the structural advantages of the materials with the properties of the materials will be the focus of future research.

## 5. Conclusions and Future Perspectives

Electrospinning technology has attracted more and more attention in recent years as a highly versatile technology for preparing micro-nano-level fibers with diameters. Electrospinning fiber material has an extremely high specific surface area, high porosity, adjustable fiber morphology and surface function. These characteristics make the electrospun fiber material possess a series of ideal properties, which can meet the application requirements of various fields such as biomedicine and tissue engineering. With the development of coaxial electrospinning, side-by-side electrospinning and triaxial electrospinning, high-quality nanofibers with core–sheath structure, Janus structure and triaxial structure have been prepared. In the future, more advanced and complex multi-fluid electrospinning technology can be utilized to produce nanofibers with new structures. At the same time, electrospun nanofibers can carry a variety of active substances and have the ability to continuously release drugs and nanoparticles, which is highly beneficial for improving the overall performance of the wound.

The development of wound dressings still faces some challenges that need to be considered, and many researchers have also given different views. Different wound exudates are extremely different, which makes it difficult to find the ideal dressing for all wound types [49,121]. Therefore, the bioactive nanofiber dressing produced by electrospinning technology takes into account the characteristics of the wound healing process, stimulates cell migration, and controls inflammation. At the same time, it relieves pain by releasing related drugs, and can be effectively used for acute or chronic wound treatment. However, this advanced dressing needs to be evaluated in a large number of clinical trials to ensure the final clinical application. We believe that our review provides insights and references for the further development of electrospun nanofibers in clinical applications. In the future, it is necessary to improve the performance of electrospun fibers further, realize the standardization of drug-loaded fiber preparation methods, and prepare new functional wound dressings to provide more technical support for biomedical applications.

## Figures and Tables

**Figure 1 membranes-11-00770-f001:**
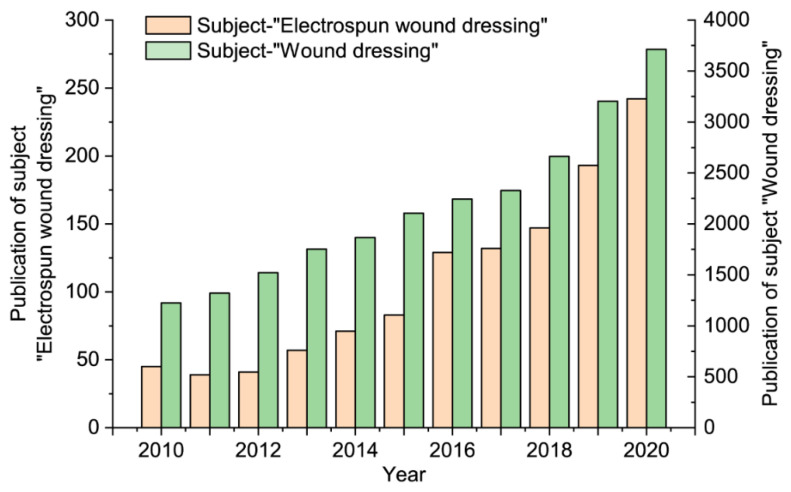
Statistics of literature retrieval on the “Web of Science” platform with the subject of “Wound dressing” and “Electrospun wound dressing”, respectively.

**Figure 2 membranes-11-00770-f002:**
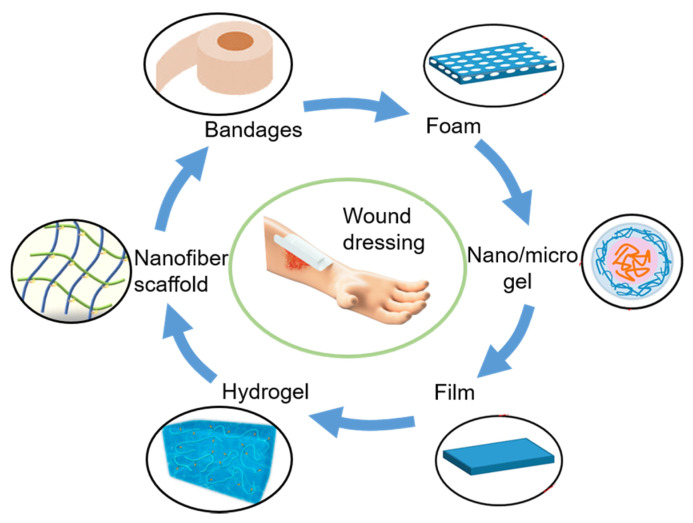
Structure of different wound dressing.

**Figure 3 membranes-11-00770-f003:**
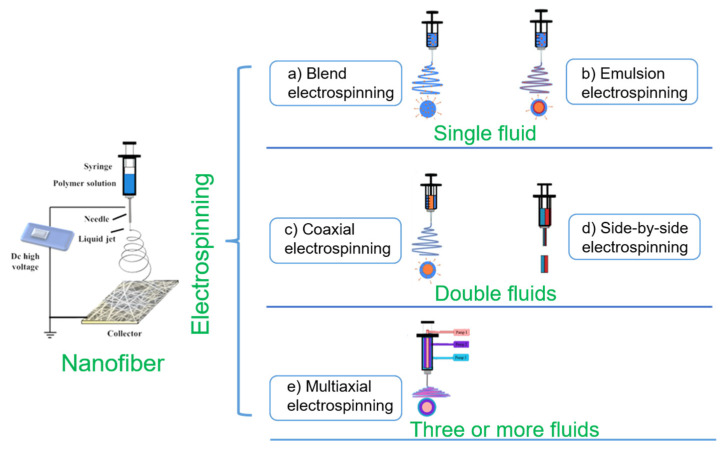
Process classification of electrospinning technology (adapted from [72], with permission from MDPI, 2021).

**Figure 4 membranes-11-00770-f004:**
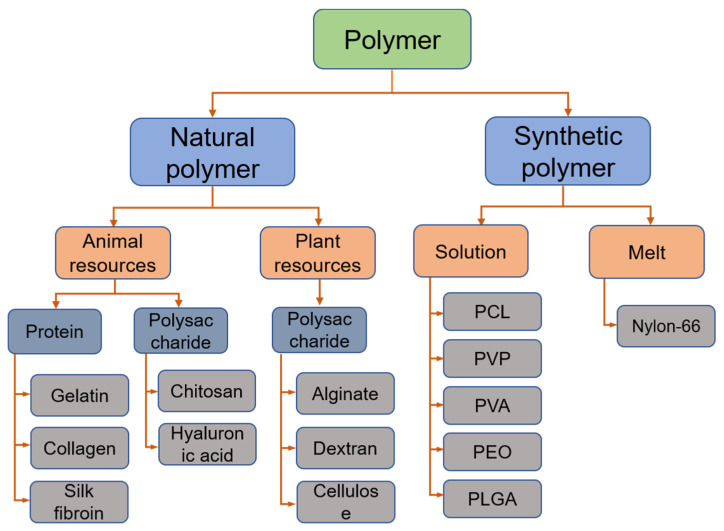
Common polymers used in electrospun wound dressings.

**Figure 5 membranes-11-00770-f005:**
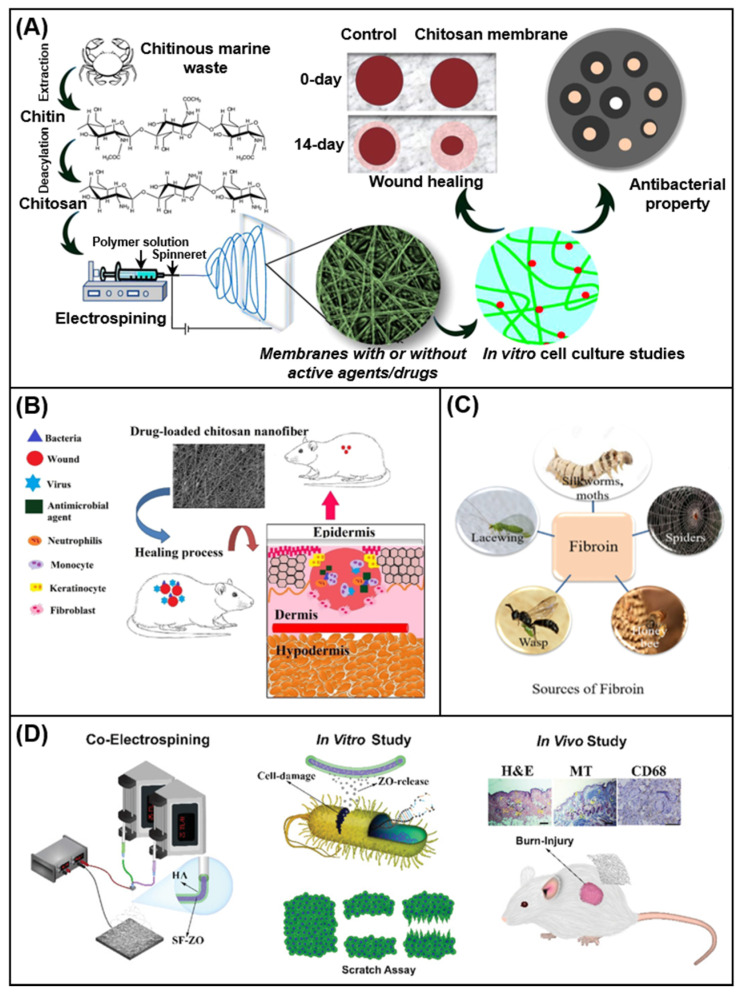
(**A**) Chitosan extracted from natural sources and developed into electrospun fiber membranes and its application in wound healing [34]; (**B**) a schematic diagram of the wound healing process of a drug-loaded chitosan dressing [103]; (**C**) the sources of fibroin [107]; (**D**) preparation and in vitro and in vivo study of HA/SF-ZO nanofiber by coaxial electrospinning [108].

**Figure 6 membranes-11-00770-f006:**
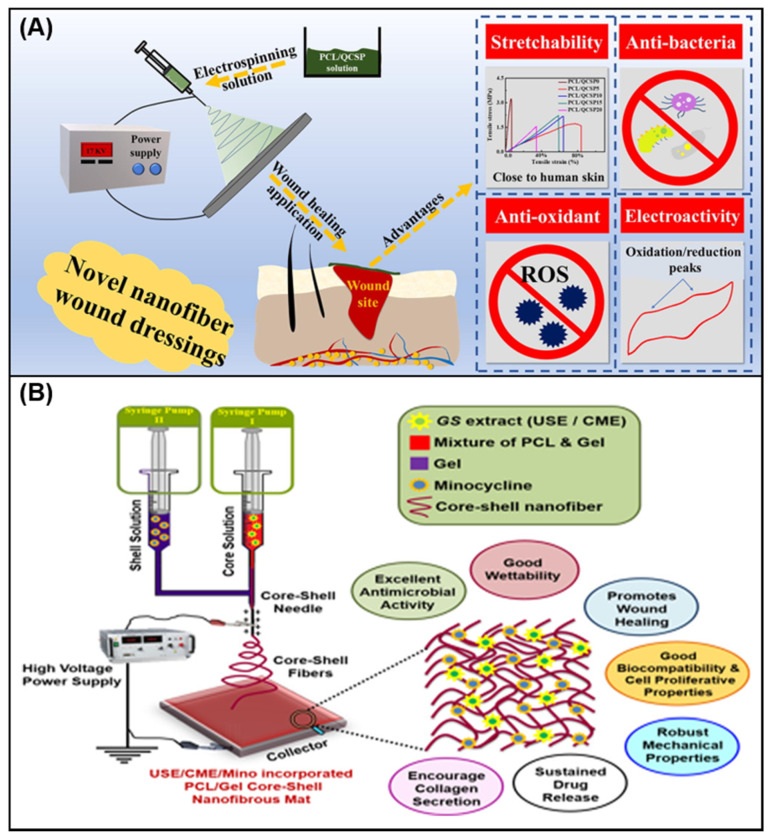
(**A**) Electrospinning PCL/QCSP nanofiber membrane with anti-bacteria, anti-oxidant, stretchability and electroactivity [113]; (**B**) preparation of coaxial electrospinning nanofiber mat and its excellent properties [117].

**Figure 7 membranes-11-00770-f007:**
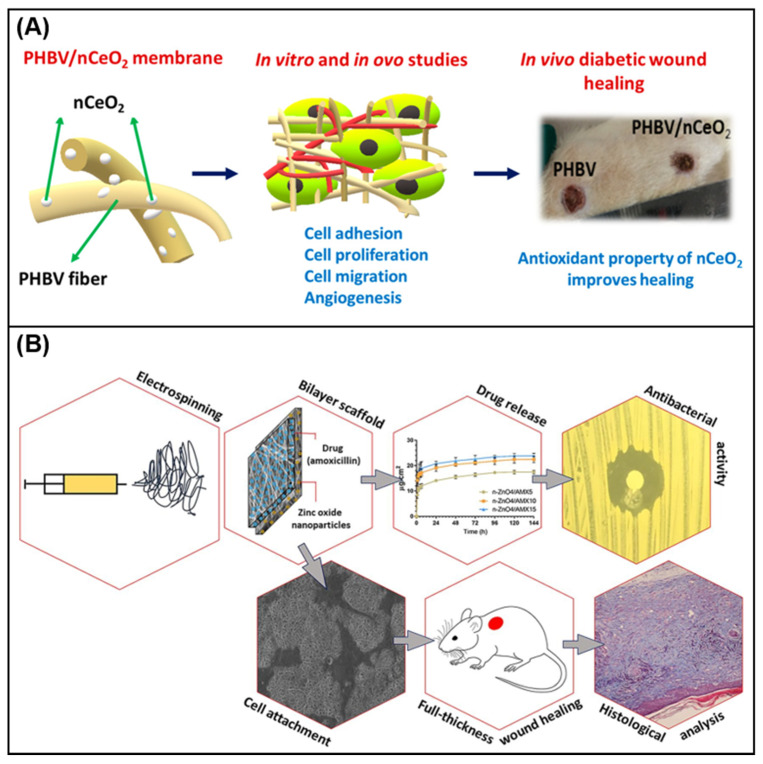
(**A**) PHBV/nCeO_2_ nanofiber membrane in cell adhesion, migration and wound healing research [124]; (**B**) the electrospun antibacterial bilayer nanofiber scaffold is used to promote the various characterization analysis of the full-thickness skin defect healing in mice [126].

**Figure 8 membranes-11-00770-f008:**
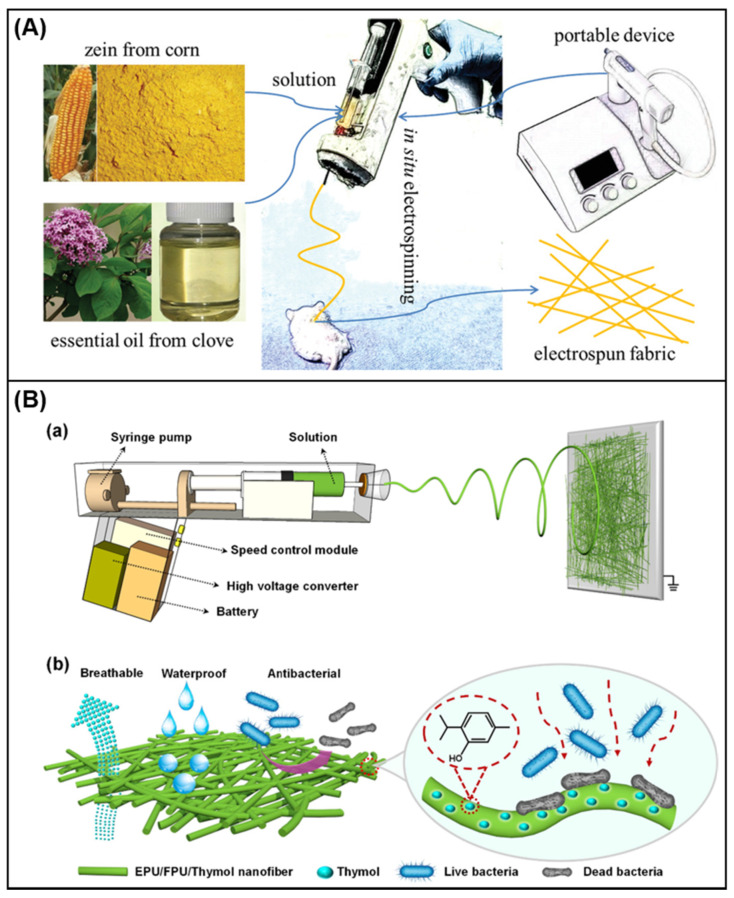
(**A**) In situ electrospinning process [148]; (**B**) [149] (**a**) schematic diagram of portable electrospinning device and preparation of EPU/FPU/thymol nanofiber; (**b**) schematic diagram of the breathable, waterproof and antibacterial functions of EPU/FPU/Thymol nanofiber.

**Figure 9 membranes-11-00770-f009:**
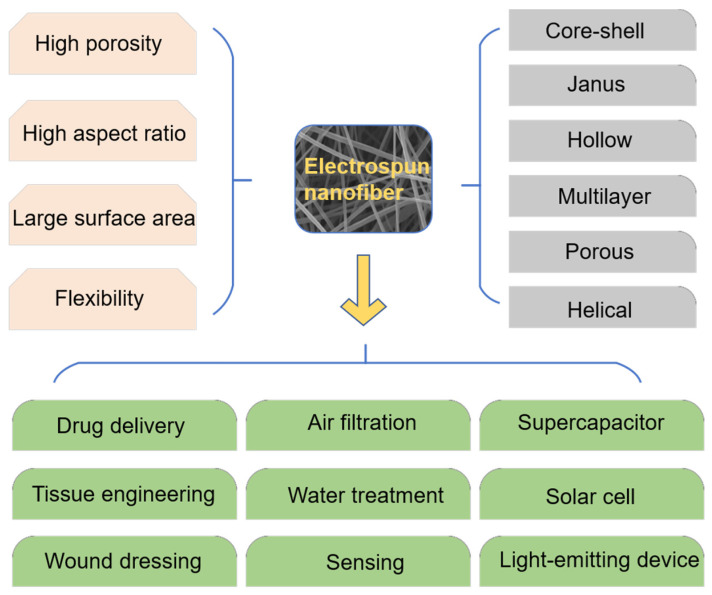
Structure, performance and application of electrospun nanofiber.

**Table 1 membranes-11-00770-t001:** Types of wound dressing.

Nature	Category	Advantages	Disadvantages	Ref.
Traditional wound dressing	Gauze, lint, bandage	Easy to use and economical	1. Dry, unable to maintain a moist healing environment 2. Adhering to the wound site is difficult to remove	[46]
Modern wound dressing	Film	1. Transparent, can observe wound changes 2. Form a bacterial barrier 3. Gas and water vapor permeability	1. Absorptive capacity is not strong 2. Obstruct the regeneration of epithelial tissue	[47]
	Foam	1. High water absorption performance to maintain the moist environment of the wound 2. Change the dressing without damage	1. Weak adhesion 2. Completely opaque	[48]
	Hydrocolloid	1. Stimulate tissue autolysis and debridement 2. The closed structure blocks the invasion of external bacteria	1. Poor degradability 2. Produce a special smell	[49]
	Hydrogel	1. Ability to replenish water and maintain a humid environment 2. Comfortable and easy to replace	1. No adhesion, low mechanical strength 2. High water content, limited absorption capacity, not suitable for wounds with high exudate	[50]
	Alginate	1. Non-toxic, fast hemostasis 2. Good air permeability 3. Biodegradation	Not suitable for dry wounds	[51]
Bioactive wound dressing	Drug-loaded dressing, antibacterial dressing	1. Good biocompatibility 2. Anti-inflammatory and antibacterial 3. Promote the growth of cells and tissues	Induce immune response	[52]

**Table 2 membranes-11-00770-t002:** The latest literature on polymer materials and bioactive ingredients used in electrospinning to promote skin wound healing.

Scaffold Material	Additional Polymer	Bioactive Ingredients	Solvent	Electrospinning Technique	Highlights	Ref.
Gelatin	CA	Berberine	HFP	Blend	Has strong antibacterial activity and is suitable for the management and treatment of diabetic foot ulcer	[127]
	CA/PVP	Gentamicin	Acetic acid, ethanol	Bi-layer	Thermal stability, wettability characteristics and antibacterial activity	[128]
Collagen	EC/PLA	Silver sulfadiazine	Chloroform, ethanol	Blend	The antibacterial performance showed inhibitory activity against Bacillus (9.71 ± 1.15 mm) and *E. coli* (12.46 ± 1.31 mm), promoted cell proliferation and adhesion	[129]
	Zein/PCL	n-ZnO, aloe vera	Chloroform, ethanol	Blend	The developed nanofibers revealed good cell compatibility	[130]
CS	PCL	Lidocaine hydrochloride, mupirocin	HFIP, DCM	Dual	Have the functions of promoting hemostasis, antibacterial, and drug release.	[131]
	PEO/CNC	Acacia extract	Acetic acid	Blend	A continuous release of natural acacia extract from nanofibers occurred during 24 h	[132]
SF	PLGA	Artemisinin	HFIP	Blend	The fabricated membrane shows anti-inflammatory properties without cytotoxicity	[133]
	PCL/PVA	Curcumin	Formic acid, dichloromethane	Blend	Accelerate wound healing in diabetic mice	[134]
Alginate	PVA/CS	Dexpanthenol	Acetic acid	Coaxial	Not only is it non-toxic to fibroblasts, but it also has a certain effect on cell attachment and morphology	[135]
	PVA	Cardamom extract	Distilled water	Blend	Have good biocompatibility and antibacterial properties	[136]
PVP	EC	CIP, AgNP	Ethanol, acetic acid, acetone	Side-by-side	Janus fiber has good bactericidal activity	[125]
	PLA/PEO/Collagen	Cefazolin	DCM, DMF, HFIP, ethanol	Coaxial	Antibacterial studies on wounds show that they can effectively inhibit the growth of microorganisms.	[137]
PCL	CS	Aloe vera	Acetic acid	Blend	Have good antibacterial properties and biocompatibility	[138]
	CS	Curcumin	Ethanol, acetic acid	Blend	Shows antibacterial, anti-oxidant and wound healing capabilities	[139]
	Gelatin	Oregano oil	HFIP	Blend	Good biocompatibility and antibacterial activity	[140]
	/	Urtica dioica, n-ZnO	DMF, DCM	Blend	The hybrid scaffold shows high antibacterial activity and cell viability	[141]
	Gelatin	Clove essential oil	Glacial acetic acid	Blend	Antibacterial activity	[142]
PVA	CS/Starch	/	Double-distilled water, acetic acid	Blend	Proper tensile strength and elongation, excellent biocompatibility and antibacterial activity	[143]
	CS	/	Acetic acid	Blend	Good physical and chemical properties, biocompatibility and antibacterial properties	[144]
PEO	CS	Vancomycin	Acetic acid	Blend	Antibacterial effects against *S. aureus*	[145]
	CS	Teicoplanin	Acetic acid	Dual	Wound closure was significantly improved	[146]

HFP: Hexafluoropropylene, EC: Ethylcellulose, PLA: Polylactic acid, HFIP: 1,1,1,3,3,3-hexafluoro-2-propanol, DCM: Dichloromethane, PEO: Polyethene oxide, CNC: Cellulose nanocrystals, DMF: *N,N*-dimethylformamide.

## Data Availability

The data supporting the findings of this manuscript are available from the corresponding authors upon reasonable request.

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
