# Peer review of "Electrospun Medicated Nanofibers for Wound Healing: Review"

_membranes, 2021, doi:10.3390/membranes11100770_

Round 1
Reviewer 1 Report
English of this manuscript should be improved.
Author Response
Responses to Comments and Suggestions:
English of this manuscript should be improved.
We highly appreciate your precious time and efforts spent on our manuscript. Based on your valuable suggestions, we have made every effort to completely revise the manuscript. Carefully checked the language in the manuscript, modified some detailed errors and grammatical errors, and adjusted and polished some sentences. In this revised version, changes/additions to the manuscript within the document were highlighted by using red/blue text, respectively.
If there are any other modifications we could make, we would like very much to modify them and really appreciate your kind help. We hope the revised version will meet the approval for publication.
Best regards,
Dr. Xinkuan Liu
Reviewer 2 Report
Dear Authors,
This paper is well written and documented, with pertinent conclusions concerning the future of electrospun medicated nanofibers for wound healing applications.
The review is well stuctured and organized, highlighting main types of wound dressings, preparation technology of electrospun fibers (exhaustively presented), types of polymers and principal active ingredients used in electrospinning wound dressings.
Some minor English revision and spelling corrections are required.
Author Response
Responses to Comments and Suggestions:
This paper is well written and documented, with pertinent conclusions concerning the future of electrospun medicated nanofibers for wound healing applications.
The review is well stuctured and organized, highlighting main types of wound dressings, preparation technology of electrospun fibers (exhaustively presented), types of polymers and principal active ingredients used in electrospinning wound dressings.
Many thanks for these kind words!
Some minor English revision and spelling corrections are required.
We highly appreciate the valuable time and energy spent by reviewer on our manuscripts to help us improve the quality of manuscript. Thank you for your positive comments and suggestions on our paper. We have carefully checked and corrected the language problems in the text, and marked the modified places for your reference.
Best regards,
Dr. Xinkuan Liu
Reviewer 3 Report
The paper is an interesting overview on the performance of the medicated nanofibers for wound healing applications.
The topic of the manuscript is of interest for readers and the work is generally well written. The manuscript can be eventually published, but this reviewer is suggesting a major revision to address this specific concern: since there are currently multiple review articles that report recent progresses in the field, authors should clearly provide a plus, which can consist in the insertion of their critical evaluation of the state of the art, as well as in the improvement of their point of view in the perspective section.
Author Response
Responses to Comments and Suggestions:
The paper is an interesting overview on the performance of the medicated nanofibers for wound healing applications.
Thank you for your positive comments on our paper!
The topic of the manuscript is of interest for readers and the work is generally well written. The manuscript can be eventually published, but this reviewer is suggesting a major revision to address this specific concern: since there are currently multiple review articles that report recent progresses in the field, authors should clearly provide a plus, which can consist in the insertion of their critical evaluation of the state of the art, as well as in the improvement of their point of view in the perspective section.
We highly appreciate the reviewer for the precious time and efforts spent on our manuscript, and particularly, the pertinent comments and suggestions, which have helped us so much to improve the quality of our manuscript. Based on your precious suggestions, we have tried our best to revise the manuscript.
In the “5. Conclusions and Future Perspectives”, we added relevant descriptions to show our innovation, cited the latest technology for critical evaluation, and further discussed the application limitations and future development direction of electrospun nanofibers in wound dressing.The main additions are as follows:
“The development of wound dressings still faces some challenges that need to be considered, and many researchers have also given different views. Different wound exudates are very different, which makes it difficult to find the ideal dressing for all wound types [49,121]. Therefore, the bioactive nanofiber dressing produced by electrospinning technology takes into account the characteristics of the wound healing process, stimulates cell migration, and controls inflammation. At the same time, it relieves pain by releasing related drugs, and can be effectively used for acute or chronic wound treatment. However, this advanced dressing needs to be evaluated in a large number of clinical trials to ensure the final clinical application. We believe that our review provides insights and references for the further development of electrospun nanofibers in clinical applications. In the future, it is necessary to improve the performance of electrospun fibers further, realize the standardization of drug-loaded fiber preparation methods, and prepare new functional wound dressings to provide more technical support for biomedical applications. ”
Best regards,
Dr. Xinkuan Liu
Round 2
Reviewer 1 Report
It seems more acceptable now.
Reviewer 3 Report
My comments were well addressed and the manuscript modified accordingly.
I recommend publication of the paper in its current form